# FINDING STRUCTURE AND CAUSALITY IN LINEAR PROGRAMS

**Matej Zečević**[1,†]    **Florian Peter Busch**[1]    **Devendra Singh Dhami**[1,3]    **Kristian Kersting**[1,2,3]
[1]Computer Science Department, TU Darmstadt, [2]Centre for Cognitive Science, TU Darmstadt,
[3]Hessian Center for AI (hessian.AI), [†]correspondence: `matej.zecevic@tu-darmstadt.de`

## ABSTRACT

Linear Programs (LP) are celebrated widely, particularly so in machine learning where they have allowed for effectively solving probabilistic inference tasks or imposing structure on end-to-end learning systems. Their potential might seem depleted but we propose a foundational, causal perspective that reveals intriguing *intra-* and *inter-structure relations* for LP components. We conduct a systematic, empirical investigation on general-, shortest path- and energy system LPs.

## 1    OPTIMIZATION PROBLEMS, THEIR STRUCTURE AND CAUSALITY

Linear Programs (LP) stand as the entry ticket to the literature around mathematical optimization. Their simplicity is characterized by *linear* cost and constraint vectors that provide the problem of study with an inherent *structure*. Both, the LPs simplicity and its structure have leveraged its success across many applications—also in machine learning. While classical examples involve the diet problem (Dantzig, 1990), the assignment problem (Kuhn, 1955), or the shortest path (SP) problem (Bellman, 1958), more modern applications of LPs involve *probabilistic reasoning* as in MAP inference (Weiss et al., 2007). The classical problems are special instances of LPs that assume a certain structure within their constraints $\mathbf{A}$ and decision variable $\mathbf{x}$. For instance in SP the decision $\mathbf{x}$ consists of indicators to path segments $(i, j)$ to be chosen for the final path $a \to \ldots i \to j \to \ldots b$ while $\mathbf{A}$ will dictate what constitutes a "legal" path. As another example, in the highly non-trivial MAP inference case we will act on a polytope of rather abstract *objects* while inducing no special structure on the problem itself. While MAP inference is a concrete example of an important machine learning (ML) task that we might be interested in and lends itself to an LP description, the recent times have shown an increased interest by the ML community in general for *combinatorial* problems. In research around adversarial examples, LPs and its integral variants were used to evaluate robustness of trained classifier in a defensive (Tjeng et al., 2019) and active setting (Wu et al., 2020). In research around end-to-end learning systems, LPs have been made differentiable through the use of perturbations in the Gumbel-Max sense (Berthet et al., 2020). More recently, both adversarials and the differentiable LPs were combined to define a new notion of adversarial attack which directly applies to LPs (and not arbitrary classifiers) by bridging the gap to Pearlian *causality* and its hidden confounders (Zečević et al., 2021). Furthermore, a magnitude of efforts has been concerned with allowing learned agents to incorporate structured knowledge, in the constraint-sense that LPs but more general cone programs (like quadratic or semi-definite programs), to leverage performance on downstream tasks (Agrawal et al., 2019; Mandi & Guns, 2020; Paulus et al., 2021).

While it might seem that research around LPs has depleted all which LPs could possibly offer, we believe to show in this work that there is a side to LPs that has remained hidden and is yet to be discovered. We have seen that LPs provide an inherent structure through their constraints but also through the way the *semantics* of the variables is decided (e.g. the meaning of the decision variable $\mathbf{x}$). But what about the *intra-* and *inter-structure* of the LP, that is, the relations between say different $(x_i, x_j)$ or even between $(x_k, \mathbf{A}_{l,:})$? Can we identify a notion of *causality* within LPs using the formal, counterfactual theory provided by (Pearl, 2009)? While pure curiosity and the success of LPs on a wide variety of seemingly different tasks (involving ML) might not allow for a proper justification of the aforementioned research questions, an intuition lies in the inherent structure provided by LPs in that their constraints commonly carry meaning just as the variables of interest do—they are governed by certain *rules*, which one might equate to structural equations in a Structural Causal Model (SCM). An important case in support of this anticipation is omnipresent in research around sustainable energy modelling. Energy systems researcher might model certain

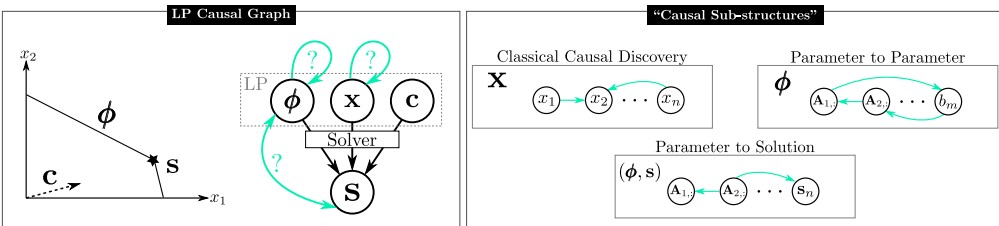

Figure 1: **Causal Perspective onto LPs.** Left, a causal graph for the constraints $\phi$, the cost $\mathbf{c}$, the decision $\mathbf{x}$ and the optimal solution $\mathbf{s}$. The "structural equation" $\mathbf{s} = f(\mathbf{x}; \phi, \mathbf{c})$ is well-known as the *solver*. But what about intra- and inter-structure (green)? Right, shows such sub-structures where classical discovery involves $\mathbf{x}$ only and LP-related causality $\phi$ or $(\phi, \mathbf{s})$. (Best viewed in color.)

physical laws like the law of conservation of energy with constraints in an LP where the decisions $x_i$ will now involve the capacity of the photo-voltaic system or the market bought electricity for a single household, and then the question arises on how these $x_i$ are causally related to each other but also to the laws and constraints that govern them (Schaber et al., 2012). While this example corroborates our previous justification, the example itself does not come to a surprise since causality (as a concept not just a specific formal notion) stands at the core of human cognition and transitively at the core of science, engineering, business, and law (Penn & Povinelli, 2007). Developmental psychology has shown children to act in the "manner of a scientist" (Gopnik, 2012), while artificial intelligence research dreams of *automating* said manner (McCarthy & Hayes, 1981). Recent strides in cognitive science have pursued a computational model of the broad term causality (Gerstenberg et al., 2021), whereas Pearlian causality specifically has pressured forward into ML research (Bareinboim et al., 2020). Various works in ML take inspiration from the latter approach leveraging causal notions and *invariances* to drive performance (Lopez-Paz et al., 2017; Kipf et al., 2018).

## 2 BEYOND CLASSICAL CAUSAL DISCOVERY

Since its core formalization, Pearlian causality has seen many iterations (Pearl, 2009; Peters et al., 2017; Bongers et al., 2021) and so have methods for identifying the causal structure given data (for an overview see Eberhardt (2017)). Typically, for an SCM $\mathcal{M} := \langle \mathbf{U}, \mathbf{V}, \mathcal{F}, P(\mathbf{U}) \rangle$ with exogenous "nature" terms $\mathbf{U}$, its joint distribution $P(\mathbf{U})$, the endogenous nodes of interest $\mathbf{V}$ and structural equations of the form $v_i \leftarrow f_i(\mathrm{pa}_i, u_i) \in \mathcal{F}$, we are classically interested in relations on $\mathbf{V}$ and sometimes also with regards to relations arising from $\mathbf{U}$ that might include hidden confounder as in non-Markovian SCM. As we motivated in the previous section, *if we know that our data stems from an LP*, then we can further *partition* $\mathbf{V}$ into sub-categories such as decision or input variable $\mathbf{x}$, parameter variables like the constraints $\phi$ (includes $\mathbf{A}, \mathbf{b}$) and the cost $\mathbf{c}$, but also the optimal solution $\mathbf{s}$. The last might be expressed as a structural equation where the former act as causes and typically we know how to compute this equation since it corresponds to the *solver*. Fig.1 (left) illustrates such a graph. Now, we might go a step further and use this *prior knowledge* (or bias) on the role of certain elements in $\mathbf{V}$ to ask questions about the relations of *parameter to parameter* or *parameter to the solution*[1], see Fig.1 (right).

To investigate these LP-specific relations we just described, we propose a data-driven approach. For this, we do not need to assume any specific knowledge on the actual LP—we simply observe data stemming from the LP. For interpretation we will assume to know what certain dimensions should refer to (e.g. feature $k$ corresponding to constraint entry $A_{i,j}$). Consider the following example which is a simple instance of the diet problem where we try to find a cost-efficient plan for consuming pizza ($x_1$) and salad ($x_2$) subject to what defines a diet to be "healthy", for instance

$$\mathbf{A}\mathbf{x} \le \mathbf{b} := (-1) \begin{pmatrix} 5 & 20 \\ 1000 & 250 \end{pmatrix} \begin{pmatrix} x_1 \\ x_2 \end{pmatrix} \ge (-1) \begin{pmatrix} 30 \\ 1800 \end{pmatrix},$$

where $\mathbf{A}_{1,:}$ denotes the amount of fiber in g (per 100 g) and $\mathbf{A}_{2,:}$ denotes the calories in kcal (per 100 g), respectively per dish, and $b_1, b_2$ denote the minimal requirements in fiber and calories (the $-1$ assures the lower bound). We can now define an indicator function $\phi(\mathbf{x})$ which is 1 if $\mathbf{A}\mathbf{x} \le$

---

[1]Note that the classical causal discovery setting is neatly captured by *input to input* in this case.

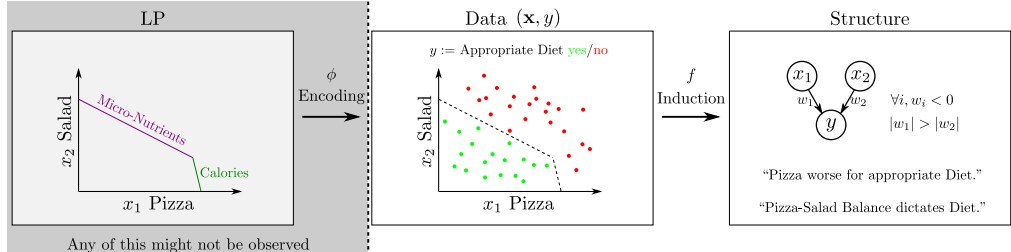

Figure 2: **Schematic View on Problem Setting.** Left, the possibly unobserved underlying LP which is being encoded into a data set. Middle, this data might consist of diets paired with "healthy-ness". Right, performing an induction reveals reconstructive properties of the original LP implicit in the data, e.g. pizza $(x_1)$ has greater influence on whether diet is healthy or not $y$. (Best viewed in color.)

$\mathbf{b}$ and 0 otherwise. Now, we might encounter our data situation where we are given a data set $\mathbf{D} := \{(\mathbf{x}_i, y_i)\}_i^n$ of size $n$ where $y_i = \phi(\mathbf{x})$ and we can assume to know that $\mathbf{x}$ refers to pizza and salad consumption whereas $y$ denotes whether the consumption is healthy or not. Given only the data and the knowledge on what it represents, what can we infer from it? Can we discover a *structural relationship*, perhaps *causal*? It turns out, applying an induction approach $f$ to the data can reveal information *implicit in this special type of data*. For our concrete example, one such resulting structure is $G : x_i \xrightarrow{w_i} y$ where $G = f(\mathbf{D})$ and $\forall i, w_i < 0$ and $|w_1| > |w_2|$. In words, we can infer 3 statements: (a) we can choose whether we consume pizza or salad *independently*, (b) our pizza and salad consumption will dictate whether our diet is healthy or not, and (c) pizza is *worse* for a healthy diet. Especially, statement (c) is interesting and highly non-trivial since it is the information about the constraints $\mathbf{A}, \mathbf{b}$ that we don't know about but that are captured implicitly by $y$. Fig.2 captures the general idea alongside this concrete example schematically.

## 3 CASE-BY-CASE EMPIRICAL INVESTIGATION

Upon establishing an intuition for strucural (causal) perspective on LPs (Sec.1) and illustrating our approach on a concrete example (Sec.2), we are now going to study the different discovery settings empirically (also including special types of LPs such as shortest path). In the following, we choose as induction method the score-based approach from (Zheng et al. (2018); abbreviated, NT) which finds the best *linear* fit of the data (under optional regularization) while satisfying an *acyclicity* constraint. Therefore, NT itself is an optimization problem[2], one that aims at finding the best weighted, directed acyclic graph (DAG) given the data. NT makes no claims on causality but it finds *correlation* (the best linear fit). This observation was recently pointed out by (Reisach et al., 2021) who showed that NT might only sort variances. Nonetheless, we choose NT for two reasons (a) correlation being an arguably appropriate proxy for an LP with its linear cost/constraints and thus pre-requisite for causation[3] at least up to confounder, and (b) NT being a simple, controllable formulation for effective experimentation. Therefore, we can expect to reveal *structure* in LPs as previously pointed out. Still, for future iterations of this work direction it would be sensible to consider the vast zoo of methods (which also provide guarantees on the causal side) and whether its majority vote keeps consistent with the subsequent results. Minor technical details can be read up in the appendix.

### 3.1 GENERAL LINEAR PROGRAMS

**Case: $\mathbf{D} = \{(\mathbf{x}_i, y_i)\}_i^n$.** If $y = 1$ for $\mathbf{Ax} \le \mathbf{b}$ , then $G = f(\mathbf{D})$ suggests the linear relationship $y = \sum_i w_i x_i$ where $w_i < 0$ (red values). Accordingly, flipping the encoding to $y = 0$ indicating feasibility we observe an expected flip in weights as well, $w_i > 0$. The $x_i$ are reported mutually independent which reflects in the data-generation where we choose $\mathbf{x}$ at random. We generally provide $f$ with 1,000 data points for our experiments. Furthermore, we observe some effects being more important to $y$ than others, e.g. increasing $x_3$ would make $y = 0$ (infeasible) the "quickest". This

---

[2]Ironically, there is a certain elegance to finding structure in an optimization problem (in the Fig.1 sense) by applying another optimization problem on top of it.

[3]This does not hold for *non-linear* settings. Absence of correlation does not imply absence of causation, while statistical dependence usually does imply the presence of causation.

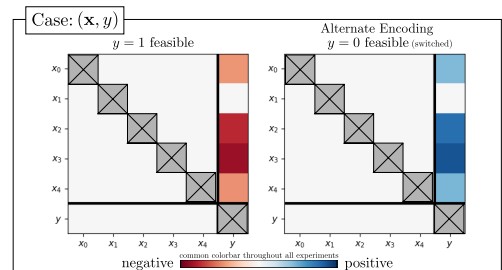 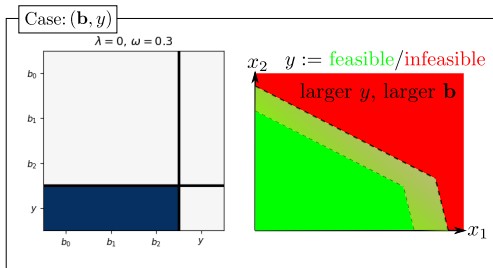

observation corroborates once more the fact that $G$ captures implicit information about constraints $\mathbf{A}, \mathbf{b}$ only from $\mathbf{D}$. The diagonals are marked out, as are any cyclic relations, since $f$ converged thus guaranteeing a DAG. The result also reflects the actual computation of $y$ although $f$ does not impose any guarantee regarding the *direction* of the fit since we could have fitted each $x_i$ via $y$.

**Case:** $(\mathbf{b}, y)$**.** In this setting we consider a family of LPs parameterized by $\mathbf{b}$ while $\mathbf{A}, \mathbf{c}$ are kept constant and $y$ is as before. We observe the same overall message as in the first case, in that $y$ will "control" the magnitude of the $b$ which dictate the placement of the polytope faces.

**Case:** $(\mathbf{b}, \mathbf{s})$**.** Again, $\mathbf{b}$ is parametric but now we consider the actual optimal solution $\mathbf{s} = \arg\max_{\mathbf{x}} \mathtt{LP}(\mathbf{x}; \mathbf{A}, \mathbf{b}, \mathbf{c})$ instead of the feasibility of an arbitrary solution. The first interesting observation (red, dotted rectangles) is $(b_1, \mathbf{s}_1) > 0$ which suggests that $b_1$ is more restrictive for obtaining a solution in that specific dimension of decision, looking at column $\mathbf{A}_{:,1}$ supports this intuition (blue denotes entry $a_{ij}>0$). The second interesting observation is the "competition" be-

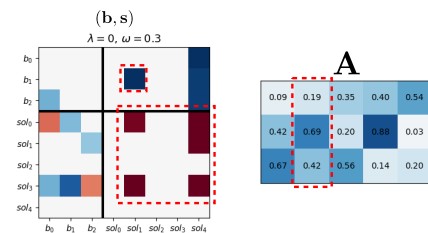

tween the dimensions of $\mathbf{s}$ since exploiting one $\mathbf{s}_i$ usually comes with a decrease in another $\mathbf{s}_j, i \neq j$.

**Case:** $(\mathbf{c}, \mathbf{s})$**.** We consider two different data sets sampled under different random seeds. We observe matching tendencies for pairs on the main diagonal $(\mathbf{c}_i, \mathbf{s}_i) > 0$ suggesting that a decrease in cost comes with an increase of selecting that dimension for the final decision. Again, confirmation

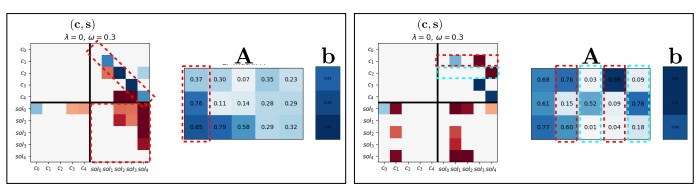

for the competition between $\mathbf{s}_i$. More interestingly, on the right data set we observe a *pattern* which requires some detailed inspection. Inspecting the row $(c_1, \mathbf{s})$ and $(c_2, \mathbf{s})$ they show a negative impact on the latter element of $\mathbf{s}$ shifted relative to each other. Inspecting the columns of $\mathbf{A}$ reveals the same patterns suggesting that for e.g. $(c_1, \mathbf{s}_1)$ the constraints were already "exhausted" such that $\mathbf{s}_3$ is being negatively impacted—and the same holds for the relations between $c_2, \mathbf{s}_2, \mathbf{s}_4$.

**Case:** $(\mathbf{c}, \mathbf{A}, \mathbf{b}, \mathbf{s})$**.** Finally, we inspect the general LP case with all defining elements being parametric. The first key observation is that all the detected relations are exclusive to *parameter and solution*. Generally, the extracted patterns confirm the results from the previous cases. A new scenario is revealed by the rows in $\mathbf{A}$ and their relation to $\mathbf{s}$ which form repeating, negative diago-

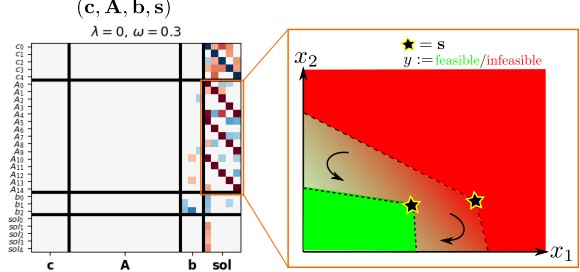

nals that suggest a shrinking of the LP polytope, thus the smaller $\mathbf{A}_{i,:}$ the smaller $\mathbf{s}_i, \forall i$.

## 3.2 Shortest Path Linear Programs

In the previous section we considered general LPs, but now we move onto the shortest path (SP) problem which is an integral variant of the LP—one of the classical guises we discussed at the beginning in Sec.1. Specifically, we consider two cases of SP much in the same fashion as before.

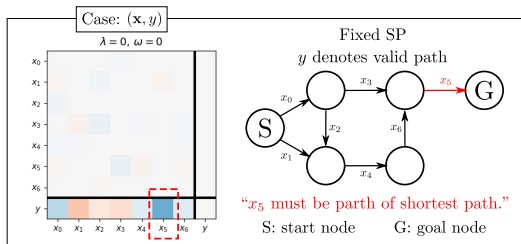 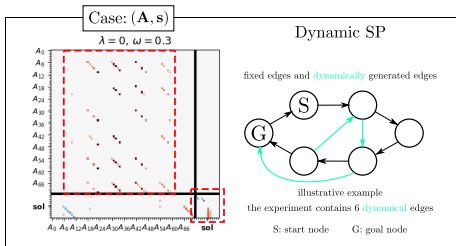

**Case:** $(\mathbf{x}, y)$. In this setting the graph to the corresponding SP problem is fixed. As before, we observe a linear relationship between the selected edges (or path segments) $x_i$ and the validity of a path (or collection of such edges) on the route from the left most node to the right most node, $y$. More interestingly, we observe a maximal activation in $x_5$ which suggests that this specific segment imposes the strongest influence on what defines a valid path. Indeed, looking at the graph we observe that if a path $\mathbf{p}$ is valid, then $x_5 \in \mathbf{p}$.

**Case:** $(\mathbf{A}, \mathbf{s})$. In addition to fixed edges, this setting also contains some dynamically generated edges. Our first observation is the pattern within $\mathbf{A}$ which simply reflects the "laws" that are characteristic of the SP problem formulation, aspects like "a visited node has exactly one incoming and at most one outgoing edge." The second observation concerns $\mathbf{s}$ where we observe that certain path segments tend to occur together in the sense that they usually belong to a valid, if not optimal, path.

### 3.3 Energy System LP of a Single Family Household

To conclude our empirical section, we consider one final LP setting in which we take inspiration from (Sch- aber et al., 2012) for a sim- ple, yet large energy system modelling the yearly energy consumption of single family

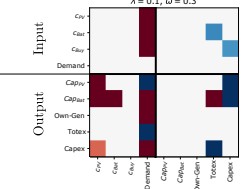 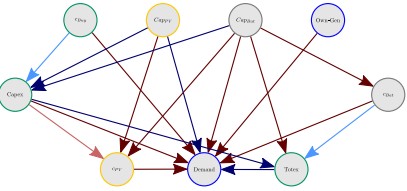

household *per hour* ($365 * 24 = 8,760$ hours multiplied by the number of constraints that are time-dependent resulting in an LP with more than $35,000$ constraints). The constraints of the LP model aspects such as energy balance, photo-voltaics production limit or battery equalities (the specific LP is listed in the appendix). Input denotes aspects of the energy system ($\mathbf{c}$), whereas output denotes the resulting optimal decisions ($\mathbf{s}$). Therefore, the investigated case corresponds to ($\mathbf{c}, \mathbf{x}$) from before. We observe *Demand* to correlate with all output dimensions, which seems appropriate. However, we also miss out on relations we'd expect to observe like for instance $Cap_{PV}$ onto $Cap_{Bat}$.

## 4 Conclusions and Future Work

Starting our discussion from the success of LPs in various applications ranging from classic ones (like diet-, assignment-, or shortest path problems) to machine learning tasks (like MAP inference or adversarial examples), we took a causal perspective and devised a new paradigm for *structural induction* from data originated in a (usually hidden) LP to reconstruct insights. Opposed to classical discovery, we can now reason about *parameter to parameter* or *parameter to solution* relations. We followed this perspective with a thorough empirical investigation on a case-by-case basis.

As pointed out in Sec.1, causality stands at the core of cognition, and transitively at the core of artificial intelligence. A counterfactual theory like the Pearlian notion to causality suggests to be an appropriate formal candidate for this endeavour. In this short work, we investigated a causal perspective but only *correlations* empirically, therefore, we propose that future research shall consider to investigate *an SCM-type of perspective on LPs theoretically*. Furthermore, raising the potential for a more general *integration and feed-back between concepts from causality and mathematical optimization* through an extension of the proposed setting to greater, real-world based examples might suggest to be fruitful.

ACKNOWLEDGMENTS

The authors thank the anonymous reviewers for their valuable feedback. This work was supported by the ICT-48 Network of AI Research Excellence Center "TAILOR" (EU Horizon 2020, GA No 952215), the Nexplore Collaboration Lab "AI in Construction" (AICO) and by the Federal Ministry of Education and Research (BMBF; project "PlexPlain", FKZ 01IS19081). It benefited from the Hessian research priority programme LOEWE within the project WhiteBox and the HMWK cluster project "The Third Wave of AI" (3AI).

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

## A   APPENDIX TO "FINDING STRUCTURE AND CAUSALITY IN LINEAR PROGRAMS"

We provide our code for public access and reproduction at: https://github.com/zecevic-matej/Finding-Structure-and-Causality-in-Linear-Programs.

The hyperparameters of the induction method are explained in Fig.3.

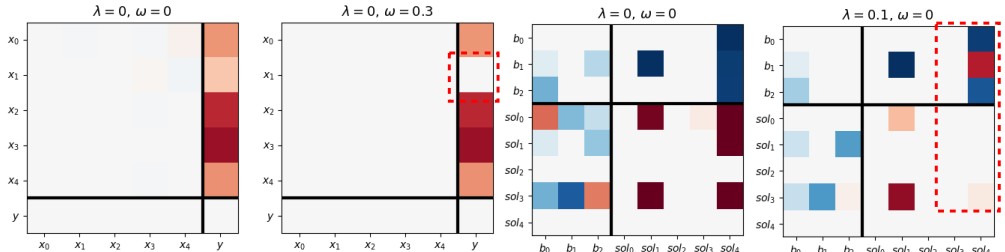

Figure 3: **Hyperparameters of Induction Method.** We deploy NT (Zheng et al., 2018) which allows for two hyperparameters $\lambda$ and $w$. While $\lambda$ controls the $L_1$-regularization terms, $w$ is a threshold for discarding low activations. Therefore, an increase in $w$ will delete signal, whereas an increase in $\lambda$ will generally reduce the *absolute* value of activations and through this also lead to an increase of activations $a$ to be discarded, $a < w$.

All experiments were conducted on a MacBook Pro (13-inch, 2020, Four Thunderbolt 3 ports) laptop running a 2,3 GHz Quad-Core Intel Core i7 CPU with a 16 GB 3733 MHz LPDDR4X RAM on time scales ranging from a few seconds (small LPs) to up to a few hours (large LPs).

The energy system LP for the single family household is shown in Tab.1.

$$
\begin{aligned}
\min_{Cap,p} \quad & c_{PV} \times Cap_{PV} + c_{Bat} \times Cap_{Bat}^S + \sum_t c_{Ele} \times p_{Ele}(t) + \sum_t c_{Gas} \times p_{Gas}(t) \\
s.t. \quad & p_{Ele}(t) + p_{PV}(t) + p_{Bat}^{out}(t) - p_{Bat}^{in}(t) + p_{Gas}(t) = D(t), \forall t \\
& p_{Bat}^S(t) = p_{Bat}^S(t-1) + p_{Bat}^{out}(t) - p_{Bat}^{in}(t), t \in 2, \ldots, T \\
& 0 \leq p_{PV}(t) \leq Cap_{PV} \times avail_{PV}(t) \times \delta t, \forall t \\
& 0 \leq p_{Bat}^{in}(t), p_{Bat}^{out}(t) \leq Cap_{Bat}, \forall t \\
& 0 \leq p_{Gas}(t) \leq U_{Gas}, \forall t \\
& p_{Bat}^S(0) = 0 \\
& 0 \leq p_{Ele}
\end{aligned}
$$

Table 1: **Energy Systems LP for Single Family Household.** A large LP that unrolls for 8760 time steps (8760 hours = 1 year). Model based on (Schaber et al., 2012), the quantities represent: Cost for Photovoltaics $c_{PV}$ (€/kW), Battery $c_{Bat}$ (€/kWh), Market Electricity $c_{Ele}$ (€/kWh), Gas $c_{Gas}$ (€/kWh), and the total Demand $D$ (kWh/Year).

