# OpenReview forum: "Finding Structure and Causality in Linear Programs"
_ICLR.cc/2022/Workshop/OSC — ICLR2022 OSC  Poster_

### Official Review · Reviewer_9Lmy · 2022-03-05
**Interesting work on relating LPs and causality**

**Rating:** 2
**Confidence:** 3

**Review:**

This paper discusses the relations between Linear Programs, such as the shortest path problem, and causality. More specifically, the paper aims at finding relations between parts of the solutions (e.g. what does including the edge 1->2 imply for the edge 2->3), the solution and constraints, and more. The correlations are found by applying the NOTEARS algorithm to sampled data from the LP, and various experimental settings are discussed.

The paper is very well written and is easily accessible to a wide audience of this workshop. The idea of relating LPs to causality is novel as far as I know, and is an interesting question. While some relations might be obvious in a small experimental setting (e.g. the pizza example in Sec 2), it is yet appealing to discuss on a larger scale. Overall, the paper does a good job in sparking an idea and giving the reader food for thought regarding the relation between LPs and causality.

The authors are also fair in stating that their current findings rely on correlations instead of causal relations. A key question for future work would be to define what it means that A 'causes' B in an LP. For instance, all parts of the solution will likely be needed to evaluate whether a solution is valid or not. For the possible solutions, you might find that some parts of the solution are independent of each other (e.g. finding the shortest path if the graph can be separated into two with a single edge), but the question remains what an orientation would imply in this setting (A -> B instead of B -> A). However, it is completely fair to leave this for future work.

Typos:
* Page 5, Line 4: the comma after 'more interestingly' is misplaced, or alternatively 'More interestingly, the maximal activation in x_5 suggests...'

---

### Official Review · Reviewer_zKUz · 2022-03-10
**Creative idea, but "causality" not shown & linear Bayes Net inappropriate**

**Rating:** 2
**Confidence:** 3

**Review:**

The paper proposes to build a linear bayesian network on the output of linear programs with the NOTEARS method. A linear program takes as input a convex polytope of constraints and a linear cost function and outputs a point in the polytope that minimizes the cost. For example, for a fixed polytope in 4 dims, for a distribution of cost functions, the authors generate a dataset of samples of the 4D cost vector and 4D corresponding solution. They then fit a linear Bayes Net to explain the 8D random vector. The authors find  some relationship between the learned Bayes Net coefficients and the polytope.

I think the approach is quite creative and it may shine some light on the inner structure of optimization algorithms. That is why I recommend acceptance of the paper for the workshop.

However, I find two major shortcomings in the method. First: while the authors make reference to causality, it is  unclear what causality means in this context. Why should the learned Bayes Net be a Causal Bayes Net? For example, why would one allow for the solution causing the cost function? Given an algorithm that maps inputs to outputs, can one ever assign in general any causality to that, without first postulating that the workings of the algorithm somehow are corresponding to some real causal/physical process (which the authors do not) or define a notion of intervention? The authors do note that they measure mere correlation, but using the word causality warrants a better explanation.

Secondly, in spite of the name, the solutions of linear programs do not depend  linearly  on the polytope or the cost function. Fitting a linear Bayes Net thus seems  inappropriate. This is  most obvious when the authors attempt to fit a linear classifier on the boundary of a convex polytope in fig 2.

---

### Decision · Program_Chairs · 2022-03-19

Accept (Poster)